# Long-Term Outcomes and Evaluation of Hepatocellular Carcinoma Recurrence after Hepatitis C Virus Eradication by Direct-Acting Antiviral Treatment: All Kagawa Liver Disease Group (AKLDG) Study

**DOI:** 10.3390/cancers13092257

**Published:** 2021-05-08

**Authors:** Joji Tani, Tomonori Senoh, Akio Moriya, Chikara Ogawa, Akihiro Deguchi, Teppei Sakamoto, Kei Takuma, Mai Nakahara, Kyoko Oura, Tomoko Tadokoro, Shima Mimura, Koji Fujita, Hirohito Yoneyama, Hideki Kobara, Asahiro Morishita, Takashi Himoto, Akemi Tsutsui, Takuya Nagano, Koichi Takaguchi, Tsutomu Masaki

**Affiliations:** 1Department of Gastroenterology and Neurology, Faculty of Medicine, Kagawa University, Kagawa 761-0793, Japan; k-takuma@med.kagawa-u.ac.jp (K.T.); m-nakahara@med.kagawa-u.ac.jp (M.N.); kyoko_oura@med.kagawa-u.ac.jp (K.O.); t-nishioka@med.kagawa-u.ac.jp (T.T.); shima@med.kagawa-u.ac.jp (S.M.); 92m7v9@med.kagawa-u.ac.jp (K.F.); hyoneyam@med.kagawa-u.ac.jp (H.Y.); kobara@med.kagawa-u.ac.jp (H.K.); asahiro@med.kagawa-u.ac.jp (A.M.); tmasaki@med.kagawa-u.ac.jp (T.M.); 2Department of Hepatology, Kagawa Prefectural Central Hospital, Kagawa 760-8557, Japan; t-senoo@chp-kagawa.jp (T.S.); a-tsutsui@chp-kagawa.jp (A.T.); t-nagano@chp-kagawa.jp (T.N.); k.takaguchi@chp-kagawa.jp (K.T.); 3Department of Gastroenterology, Mitoyo General Hospital, Kagawa 769-1695, Japan; moriyaakio@gmail.com; 4Department of Gastroenterology and Hepatology, Takamatsu Red Cross Hospital, Kagawa 760-0017, Japan; chikara.ogawa.19721202@gmail.com; 5Department of Gastroenterology, Kagawa Rosai Hospital, Kagawa 763-8502, Japan; dboy.aki0829haru1018@gmail.com; 6Department of Internal Medicine, Yashima General Hospital, Kagawa 761-0816, Japan; t.sakamoto@yashima-hp.com; 7Department of Medical Technology, Kagawa Prefectural University of Health Sciences, Kagawa 761-0123, Japan; himoto@chs.pref.kagawa.jp

**Keywords:** direct-acting antivirals, hepatocellular carcinoma recurrence, hepatitis C virus, alpha-fetoprotein, transcatheter arterial chemoembolization

## Abstract

**Simple Summary:**

In this study, the rate of hepatocellular carcinoma (HCC) recurrence (HCC-R) after hepatitis C virus eradication by direct-acting antiviral (DAA) treatment was relatively high, corresponding to 63.8% (83/130) of the studied cases. The patients who electively received palliative treatment before DAA treatment and showed a shorter interval period between the last treatment for HCC and DAA initiation, also displayed a significant increase in alpha-fetoprotein during follow-up after therapy, relative to cases without HCC-R. Overall, the survival was comparable in both groups because of the improvement in liver function tests after the DAA therapy, and thus no differences in survival rates were observed between patients with or without HCC-R. The results of this study indicate that interferon (IFN)-free DAA treatment after treatment for HCC could be recommended to improve prognosis in this subset of cases. However, it remains imperative to observe the timing, i.e., at least 9–12 months after the last treatment for HCC, before proposing DAA therapy.

**Abstract:**

There are limited studies that have evaluated the long-term outcomes in patients with hepatocellular carcinoma (HCC) recurrence after direct-acting antiviral (DAA) treatment. In this retrospective study, we aimed to investigate the recurrence rates, recurrence factors, and prognosis of 130 patients who were treated with IFN-free DAA treatment after treatment for HCC. The median observation time was 41 ± 13.9 months after DAA treatment. The recurrence rates of HCC were 23.2%, 32.5%, 46.3%, and 59.4% at 6, 12, 24, and 36 months, respectively. A multivariate analysis showed that palliative treatment prior to DAA treatment (HR = 3.974, 95% CI 1.924–8.207, *p* = 0.0006) and alpha-fetoprotein at sustained virological response 12 (HR = 1.048, 95% CI 1.016–1.077, *p* = 0.0046) were associated with independent factors for HCC recurrence (HCC-R). The 12-, 24-, and 36-month overall survival rates were 97.6%, 94.0%, and 89.8%, respectively. The 12-, 24-, and 36-month survival rates of the non-recurrence and recurrence groups were 97.7%, 97.7%, and 94.1% and 97.6%, 92.3%, and 87.9%, respectively (*p* = 0.3404). The size of the main tumor lesion and the serological data were significantly improved at the time of HCC-R after DAA treatment. This study showed an improved prognosis regardless of recurrence rate, which suggests that DAA treatment in HCV patients should be considered.

## 1. Introduction

Hepatocellular carcinoma (HCC) is the most common type of liver cancer and the fourth leading cause of cancer death worldwide [1]. Hepatitis C virus (HCV) infection is a risk factor for liver cirrhosis (LC) and HCC [2]. Therefore, suppression of HCV is important, and HCV treatment should be actively introduced. Interferon (IFN)/ribavirin combination therapy for hepatitis C has been the standard of care in the past, but the cure rate has been unsatisfactory, at approximately 50% [3,4,5]. Additionally, IFN is not suitable for older or LC patients with a low platelet count because of adverse events such as high fever, general malaise, depression, interstitial pneumonia, and decreased platelet count [6]. Therefore, the number of patients who can benefit from IFN treatment is small. In recent years, direct-acting antiviral (DAA) agents have been developed for HCV infection, and the cure rate of the current IFN-free DAA treatment (DAA) has dramatically improved to over 95%. Furthermore, this treatment can be used safely in patients with HCV [7,8,9,10,11,12].

The risk of recurrence in patients with HCC is much higher than in those with other malignancies and is associated with lower survival rates [13]. Observational studies of HCV-infected patients have shown that patients who achieve a sustained virologic response after treatment with IFN or IFN-free DAA treatment have a reduced risk of HCC, liver disease complications, and death [14,15,16]. Several studies suggested a lower rate of HCC recurrence (HCC-R) after IFN-free DAA tratment, while others reported the converse [17,18,19]. This discrepancy has not yet been resolved. 

We conducted a multicenter, longitudinal, 41-month study, namely All Kagawa Liver Disease Group Study, to retrospectively evaluate the incidence, characteristics, and predictors of HCC-R in a large cohort of HCV patients who were treated with IFN-free DAAs.

## 2. Results

### 2.1. Patients’ Characteristics

A total of 1519 patients started DAA treatment during the study period and 130 patients were enrolled in the study after applying the exclusion criteria. Patients’ baseline characteristics are shown in Table 1. Eighty-three male and 47 female patients were enrolled. The median patient age at the start of DAA treatment was 75.5 years. Thirty-three patients (25.4%) had a history of IFN-based treatment. The numbers of chronic hepatitis (CH) and LC cases were 44 and 86, respectively.

### 2.2. Cumulative Recurrence Rate of HCC

The median observation time was 41 ± 13.9 months at the end of DAA treatment. During this study, HCC recurred in 83 cases. The recurrence rate of HCC was 23.2%, 32.5%, 41.5%, 46.3%, and 59.4% at 6, 12, 18, 24, and 36 months, respectively, after DAA treatment (Figure 1). The number of patients traced to each time point is shown in Figure 1.

### 2.3. Comparison of Patient Characteristics between Cases with and without HCC-R

The results of the univariate analysis between cases with and without HCC-R are shown in Table 2. No significant differences in sex, age, HCV genotype, body mass index (BMI), history of diabetes mellitus (DM), DAA regimen, or history of IFN treatment were identified. Among the tumor-associated factors, there were no significant differences in the number of tumors or the maximum tumor diameter. At the time of DAA administration, no significant differences were observed in alanine aminotransferase, aspartate aminotransferase, bilirubin, albumin, platelet count, α-fetoprotein (AFP), des-γ-carboxyprothrombin (DCP), total cholesterol, fibrosis-4 (FIB-4), aspartate aminotransferase (AST)/aspartate aminotransferase-to-platelet ratio index (APRI), Wisteria floribunda agglutinin (WFA)-positive Mac-2 binding protein (M2BP), albumin–bilirubin (ALBI) score, or Child–Pugh score. Significant differences were observed in the interval period between the last treatment for HCC and the initiation of DAA treatment, the number of treatments for HCC, and palliative treatment (i.e., transcatheter arterial chemoembolization (TACE) + molecular targeted agent (MTA)) for HCC before the DAAs (*p* = 0.0221, *p* = 0.0071, *p* = 0.0288, *p* = 0.0058, and *p* = 0.0208, respectively).

### 2.4. Risk Factors for HCC-R after DAA Treatment

Among DAA-treated patients, the univariate analysis identified the following risk factors for HCC-R: an interval period between curative treatment for HCC and initiation of the DAA treatment (*p* = 0.0288), total number of treatments (*p* = 0.0058), palliative treatment prior to DAA treatment (*p* = 0.0288), total bilirubin at SVR12 (*p* = 0.0157), AFP at SVR12 (*p* = 0.0028), total cholesterol at SVR12 (*p* = 0.0134), and ALBI score at SVR12 (*p* = 0.0258) (Table 2). A Cox hazard analysis was performed for the interval between curative treatment for HCC, total number of treatments, palliative treatment prior to DAA, total bilirubin at SVR12, AFP at SVR12, total cholesterol at SVR12, and ALBI score at SVR12. Palliative treatment prior to DAA treatment (hazard ratio (HR) = 3.974, 95% confidence interval (CI) 1.924–8.207, *p* = 0.0006), and AFP at SVR12 (HR = 1.048, 95% CI 1.016–1.077, *p* = 0.0047) were found to be significant risk factors contributing to HCC-R after DAA treatment (Table 3).

### 2.5. Receiver Operating Characteristic Analysis and Diagnostic Value

The serum AFP level (AFP) at SVR12 was shown to be an independent factor for HCC-R after DAA treatment. Because the AFP was analyzed as a continuous variable, the cutoff value of the AFP was used as the diagnostic value. The cutoff value of the AFP at SVR12 was set as 8.0 ng/mL, based on the receiver operating characteristic (ROC) value for HCC-R, which was calculated according to the AFP at SVR12 noted in the original set (sensitivity, 0.4861; specificity, 0.900; AUC, 0.72667).

### 2.6. Cumulative Incidence of HCC-R According to the Serum AFP Level (AFP) at SVR12 and Palliative Treatment

As shown in Figure 2b, the cumulative incidence of HCC-R in patients was significantly lower than in patients with an AFP >8 (3-year incidence: 52.7% with AFP ≤ 8 and 76.3% with AFP > 8, *p* = 0.0327, log-rank test) and palliative treatment (3-year incidence: 52.5% without TACE and 92.8% with TACE, *p* < 0.001, log-rank test).

### 2.7. Comparison of Overall Survival Time between Cases without and with HCC-R

During the follow-up period, 16 patients (12.3%) died and 114 patients (87.7%) were still alive at the end of the observation. The 12-, 24-, and 36-month overall survival rates were 97.6%, 94.0%, and 89.8%, respectively.

Among the patients without HCC-R, three patients (6.4%) died and 44 patients (93.6%) were still alive; among the cases with HCC-R, 13 patients (15.7%) died and 70 patients (84.3%) were still alive. The 12-, 24-, and 36-month survival rates of the cases without and with HCC-R were 97.7%, 97.7%, and 94.1% and 97.6%, 92.3%, and 87.9%, respectively (Figure 3). Furthermore, there was no significant difference in overall survival (OS) between the non-recurrence and recurrence groups (*p* = 0.3404, Figure 3).

### 2.8. Relative Changes in HCC Status and Serological Data at Baseline and at HCC-R after DAA Treatment

This subanalysis compared factors in the 87 cases with HCC-R at two time-points that were defined as “at baseline” and “at HCC-R after DAA treatment”. The size of the main tumor lesion was significantly smaller at HCC-R after DAA treatment as compared with the baseline (*p* = 0.0138, Table 4). In addition, AST, ALT, albumin, platelet counts, AFP, and ALBI score were significantly improved at HCC-R after DAA treatment as compared with the same figures at baseline (*p* < 0.001, *p* < 0.001, *p* < 0.001, *p* = 0.0032, *p* = 0.0459, and *p* < 0.001, respectively, Table 4).

## 3. Discussion

DAA agents have recently been introduced as simple and safe antiviral oral treatments for HCV infection. The application of DAA agents has facilitated the eradication of HCV, even in older and LC patients, and resulted in a high SVR rate for HCV infection, as well as a reduction in the incidence of HCV-associated HCC which is expected in the future. However, it has been reported that in patients with a history of HCC, the rate of HCC-R after 6 months of DAA treatment was as high as 27.6–28.8% [19,20]. In this study, we analyzed the effect of DAA treatment without IFN on HCC-R in patients who received curative treatment for HCC over a medium to long term duration. There have been several studies that have investigated the recurrence of HCC after DAA treatment in patients with previous HCC, but to the best of our knowledge, no analysis of the effect on HCC-R over a medium to long term duration has been conducted.

There are two reasons for HCC-R after DAA treatment. One reason is that HCC may have been missed by imaging diagnosis before DAA administration, and another is new HCC onset after DAA treatment. In the present study, we repeatedly reviewed all images obtained by ultrasonography (US), computed tomography (CT), or magnetic resonance imaging (MRI) to prevent any oversights, and we excluded patients with HCC before starting DAA treatment. The recurrence rates of HCC in our collaborative study were 23.2%, 32.5%, 41.5%, 46.3%, and 59.4% at 6, 12, 18, 24, and 36 months after DAA treatment, respectively. It has been reported that the risk of HCC-R after DAA treatment was similar between treated and untreated patients, and the risk of early HCC onset and recurrence after virus eradication was similar between IFN and DAA treatment [16,21,22]. The recurrence rate of HCC at 6 months after DAA treatment was in the 20% range, as previously reported [19,20]. In DAA treatment of previously treated HCC patients, the short-term elimination of HCV caused a decrease in the expression of cytokines, including endogenous IFNs, which led to the re-emergence and proliferation of HCC cells that were originally latent in the liver [23,24]. DAA treatment may create a permissive state for cancer growth. Similarly, the results of this study showed that the recurrence rates of patients treated for HCC after DAA treatment were as high as 32.5% at 12 months, 46.3% at 24 months, and 59.4% at 36 months. However, it is generally assumed that patients with a history of HCC will have microscopic HCC, and the evaluation of tumor suppression by DAA agents in these patients should be undertaken when the effects of microscopic HCC have disappeared. Furthermore, a longer observation period is still required. Because of the risk of HCC-R, surveillance at four-month intervals after DAA treatment has been recommended for patients with a history of HCC [25]. Similarly, in this report, strict follow-up after achieving a SVR was important because patients with a history of HCC had a high rate of HCC-R even after DAA treatment.

It is necessary to conduct surveillance of HCC-R after DAA treatment on the premise that the rate of recurrence will be higher than that of IFN treatment because of advanced age and hepatic fibrosis. In this situation, how to stratify the risk groups is an important research issue. The effect of DAA treatment on HCC-R after curative treatment remains unclear. Several retrospective studies have shown that the progression of liver fibrosis, tumor-specific factors at the time of curative treatment (such as number and size), number of previous treatment sessions, and the interval period between the last treatment for HCC and the DAA initiation, were associated with the rate of HCC-R after DAA treatment [26,27,28,29]. In this study, the univariate analysis identified the interval period between the last treatment for HCC and the DAA initiation, the total number of treatments, the palliative treatment before the DAA treatment, in addition to total bilirubin, AFP, total cholesterol, and ALBI score at SVR12, as risk factors. Moreover, by multivariate analysis, taking into account significant factors in the univariate analysis, palliative treatment before DAA treatment and the AFP at SVR12 were identified as independent risk factors for HCC-R.

Elevated AFP levels are recognized to be one of the common diagnostic markers for HCC, particularly when >200 ng/mL [30,31]. It is known that elimination of the virus by IFN significantly decreases AFP levels [31,32]. Thus, in our study, the AFP at pre-treatment did not appear to b significantly different between cases without and with HCC-R, but at post-treatment evaluation, this marker became an independent predictor of carcinogenesis, as also described in [33], being at cutoff values >8.0 ng/mL statistically associated with HCC-R.

In the ANRS cohort study, patients with previously treated HCC received highly curative treatment such as percutaneous ablation, hepatic resection, and liver transplantation [21]. Minami et al. reported that when the cumulative HCC recurrence rate after RFA treatment was compared among a non-treatment group, an IFN-treated group, and a DAA-treated group, the DAA-treated group did not have a higher recurrence rate [34]. In contrast, a study by Reig et al. included patients who had undergone chemoembolization, which has a high rate of liver cancer recurrence [20]. Different background factors, such as treatment methods and selection criteria for analyses, may have significant effects on the recurrence rate of HCC after DAA treatment. In the present study, many cases of chemoembolization were included, resulting in a higher recurrence rate, and chemoembolization may have been an important recurrence factor. A single-arm prospective clinical trial of TACE in 99 patients with first-line unresectable hepatocellular carcinoma showed a median time to progression of 7.8 months and a response rate of 73% at first treatment [35]. Our study also showed a median time to progression of 8.4 months from TACE before DAA treatment. DAA treatment does not prevent HCC recurrence after the most recent TACE, and TACE is a palliative treatment and not a curative treatment.

A meta-analysis demonstrated that viral eradication by IFN treatment prevents HCC-R [36]. Reig et al. suggested that DAA treatment may promote HCC-R, whereas Waziry et al. found no significant differences between DAA and IFN treatments in a meta-analysis adjusted for age and observation period [20,33]. In a study of 5192 patients with hepatocellular carcinoma, Bucci et al. showed that a wide implementation of surveillance resulted in early diagnoses with highly curative treatment, which is important for long-term survival [37]. In this study, a CT or MRI was performed three to four times a year even after HCV eradication to ensure an early diagnosis of HCC. Therefore, 83 cases had HCC-R after DAA treatment; however, these cases did not show the worsening of tumor-specific factors (such as number, size, and BCLC staging classification), of tumor markers (such as AFP and DCP) or of liver functional indexes (such as AST, ALT, and bilirubin), and of Child–Pugh score. On the contrary, they appeared to have improved from baseline. In addition, there was a significant improvement in liver functional reserve before and after DAA treatment. The guidelines recommend that maintenance of liver function is an important factor in providing effective HCC treatment [38,39]. DAA agents improve decompensated cirrhosis, which restores the liver functional reserve [40]. In the present study, the Child–Pugh score and ALBI score were significantly improved before and after DAA treatment in patients with HCC-R. As reported in a recent comprehensive review, the Child–Pugh score at the time of HCC-R affected treatment allocation and overall survival [41]. The improvement in the Child–Pugh score and ALBI score by HCV eradication after DAA treatment may contribute to an improved prognosis by enabling the selection of an appropriate treatment for the pathological condition of HCC.

It is important to examine the survival rate of patients who received DAA treatment after curative treatment for HCC, and the observation period of this study was 41 ± 13.9 months, which was a medium to long term duration. In this study, OS at 12, 24, and 36 months was good, at 97.6%, 94.0%, and 89.8%, respectively. As mentioned above, DAA treatment may be associated with an improvement or preservation of liver function. In this study, the survival rate of patients who received DAA agents after treatment for HCC was evaluated in the recurrence and non-recurrence groups. Remarkably, there was no significant difference in survival rates with or without HCC-R if hepatitis C was cured using DAA agents after HCC treatment. The prognosis of HCC patients is known to be dependent on tumor number, tumor size, and liver function [42,43]. In this study, patients with recurrent HCC had no change in tumor-specific factors (such as number, size, and BCLC staging classification) because of proper surveillance. This increase in the survival rate indicates that the improvement in liver function reduced the mortality rate due to liver failure, and indicates that the improvement in liver function has expanded the treatment options for HCC-R. Among 11 studies on HCV-related early hepatocellular carcinoma, a meta-analysis of an untreated group of HCV patients who received complete treatment showed a 24-month recurrence rate (47%) and a 36-month survival rate (79.8%) [44]. The 24-month recurrence rate after HCV eradication by DAA treatment in this study was 45.7%, and the 36-month overall survival rate was 89.8%. In particular, the 36-month survival rates of the cases without and with HCC-R were 94.1% and 87.9%, respectively. Eradication of HCV using DAA agents does not improve the recurrence rate of HCC, but overall survival may improve with or without HCC-R. Among the 83 patients with HCC-R, 13 patients died, nine from hepatic diseases including HCC-R, and four from extrahepatic diseases; among the 47 patients without HCC-R, three patients died, all from extrahepatic diseases. Lee et al. reported that anti-HCV seropositive patients had higher mortality from hepatic and extrahepatic diseases than anti-HCV seronegative patients, with multivariable-adjusted hazard ratios of 1.89 ( 95% CI 1.66–2.15) for all causes, 12.48 (95% CI 9.34–16.66) for hepatic diseases, and 1.35 (95% CI 1.15–1.57) for extrahepatic diseases [45]. DAA treatment is expected to improve the prognosis, but surveillance of extrahepatic disease, as well as hepatic disease, is important because HCV patients are at high risk of death from extrahepatic disease.

There are several limitations of this study. This was a retrospective cohort study, and there was a selection bias due to insufficient data parameters and the addition of palliative treatment. In addition, the diagnosis of HCC curability and HCC-R was not strictly centrally defined and was determined by the attending physician. However, despite these limitations, the value of this study is significant due to the fact that we examined a large number of patients treated in a real clinical setting.

## 4. Materials and Methods

### 4.1. Patients

This project was a multicenter study comprised of six institutions (Kagawa University Hospital, Kagawa Prefectural Central Hospital, Takamatsu Red Cross Hospital, Mitoyo General Hospital, Kagawa Rosai Hospital, and Yashima General Hospital) in Kagawa (i.e., All Kagawa Liver Disease Group Study), Japan. A total of 1519 patients with HCV infection who received DAA treatment between September 2014 and November 2019 were included in the present study. The study population included 763 men and 750 women. The median age of patients at the start of DAA treatment was 69 years. The present study was conducted in accordance with the guidelines of the Declaration of Helsinki and was approved by the ethics committee of Kagawa University, Faculty of Medicine (approval no. Heisei-27-174). The requirement for informed consent from the participants was waived because of the retrospective nature of the study.

The exclusion criteria were as follows: (i) patients co-infected with hepatitis B virus or human immunodeficiency virus, or patients with other liver diseases, including primary biliary cholangitis and autoimmune hepatitis; (ii) patients who did not achieve SVR12 following DAA treatment; and (iii) patients who had no HCC before DAA treatment. A total of 130 patients with a history of HCC who achieved SVR12 following DAA treatment were finally included in this study (Figure 4). The interval period between the last treatment for HCC and the DAA initiation was 6.6 ± 30.6 months. Among the 130 patients, 22 patients were treated with daclatasvir and asunaprevir for 24 weeks, 53 patients were treated with SOF + LDV for 12 weeks, 28 patients were treated with SOF and ribavirin for 12 weeks, eight patients were treated with elbasvir and grazoprevir for 12 weeks, nine patients were treated with OBV + PTV + r for 12 weeks, and the remaining 10 patients were treated with glecaprevir and pibrentasvir for 8–12 weeks. None of the patients enrolled in the present study received multiple courses of DAA treatment because of recurrence or non-response to initial treatment. Each patient was examined by CT or MRI to exclude the presence of HCC at the onset of DAA administration. Serum samples for HCV RNA measurements were collected at screening: treatment weeks 1, 2, 4, 8, and 12 (or early discontinuation) and post-treatment weeks 4, 8, 12, and 24 (or early discontinuation). HCV RNA was extracted from a 200 μL serum sample using an Qiagen mRNeasy serum-plasma kit (Qiagen, Hilden, Germany), according to the manufacturer’s instructions. Using a commercially available device, a TaqMan polymerase chain reaction method (COBAS AmpliPrep/COBAS TaqMan HCV test version 2, Roche Molecular Diagnostics, Pleasanton, CA, USA) (lower limit of quantification 1.610 IU/mL and lower limit of detection 1.210 IU/mL) was used to detect HCV RNA, according to the manufacturer’s instructions. HCV genotyping was also performed with the commercial VERSANT HCV Genotype 2.0 assay (LiPA 2.0) (Siemens HealthcareGmbH, Erlangen, Germany), and HCV sequencing for the detection of six HCV genotypes 1a, 1b, 2, and 3 in the serum samples, according to the manufacturer’s instructions.

### 4.2. HCC Surveillance after DAA Treatment

Dynamic CT or MRI was performed at the time of SVR12 and 3 to 4 times a year thereafter. CT or MRI examinations were diagnosed by experienced hepatologists or radiologists at each institution. The AFP and other serological data were measured every 3 months. These surveillance protocols are standard methods in Japan. For the diagnosis of HCC, we adopted the 2005 guidelines of the American Association for the Study of Liver Disease [46]. Typical HCC pattern imaging criteria were as follows: (i) hypervascularity was defined as focal lesion hyperattenuation relative to the liver during the arterial phase of each imaging method, and a washout appearance was observed during the portal and parenchymal phase; and (ii) tumors were revealed as having defects in the hepatobiliary phase of EOB-MRI. The tumor–node–metastasis stage was determined according to the HCC staging report in the Liver Cancer Study Group of Japan, 6th edition [47]. Diagnosis of LC was performed in accordance with clinical, experimental, abdominal US, and/or histological findings [48]. The FIB-4 index was calculated using the reported formula: (aspartate aminotransferase (AST, IU/L) × age (years)/platelet count (109/L) × alanine aminotransferase (ALT, IU/L)1/2). FIB-4 was used to clinically evaluate fibrosis before and after DAA treatment. A cutoff value of 3.25 was used for prediction of severe fibrosis [49,50]. The APRI was calculated using the reported formula: ((AST (IU/L)/upper limit of normal AST (IU/L)) × 100/platelet count (109/L)). A cutoff value of 0.7 was used for the prediction of severe fibrosis [51]. ALBI score was calculated based on a calculation from a previous report, i.e., (Log10 T-Bil (μmol/L) × 0.66 + Alb (g/L) × (−0.085)) [52].

### 4.3. Statistical Analysis

Values are expressed as the median. Univariate analyses for continuous variables were undertaken using Student’s *t*-test, paired *t*-test, and one-way ANOVA. For the analysis of categorical variables, Mann–Whitney U test, Fisher’s exact test, chi-squared test, proportional hazard model test, and Gray’s test with log-rank test results were performed. A multivariate analysis was performed using the Cox proportional hazards model and was applied only to variables that were statistically *p* < 0.05 in the univariate analysis. A survival analysis was performed using the Kaplan–Meier method. Statistical analyses were performed using JMP statistical software, version 15.0 (Windows version, SAS Institute, Cary, NC, USA). All *p*-values were derived from two-tailed tests, with *p* < 0.05 accepted as statistically significant. ROC and area under the curve values were calculated to define cutoff values for risk factors of HCC-R.

## 5. Conclusions

The HCC-R rate after HCV eradication was relatively high, and the risk factors for HCC-R were the AFP at SVR12 and palliative treatment (i.e., TACE + MTA) before DAA treatment. These patients electively received a palliative treatment before DAA treatment and showed a shorter interval period between the last treatment for HCC and DAA initiation, also displaying a significant increase in the AFP at SVR12 relative to cases without HCC-R. The OS was comparable in both groups because of the improvement in liver function tests after DAA treatment, and thus no differences in survival rates were observed between patients with or without HCC-R. The results of this study indicate that DAA treatment after treatment for HCC could be recommended to improve prognosis in this subset of cases. However, it remains imperative to observe the timing, i.e., at least 9–12 months after the last treatment for HCC, before proposing DAA treatment.

## Figures and Tables

**Figure 1 cancers-13-02257-f001:**
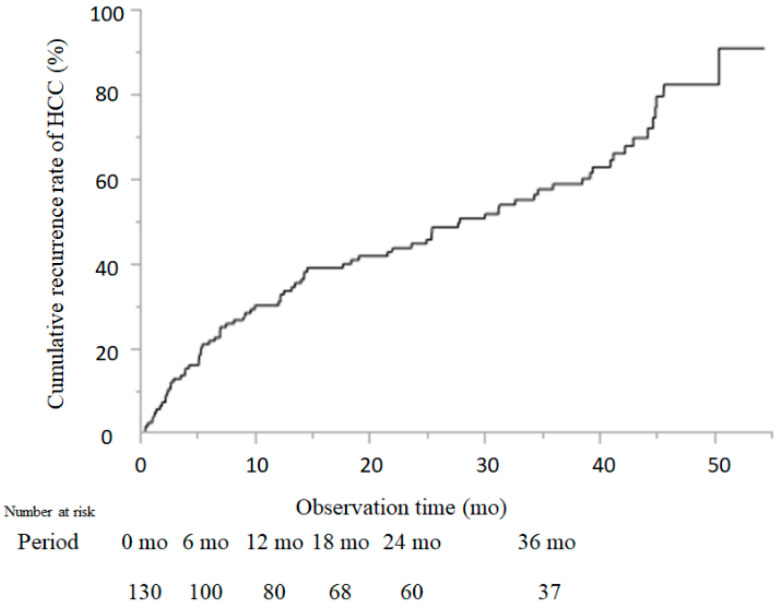
Cumulative incidence of HCC recurrence. Cumulative incidence rates of HCC recurrence are shown. The Kaplan–Meier method and log-rank test were used to assess the cumulative incidence of HCC recurrence.

**Figure 2 cancers-13-02257-f002:**
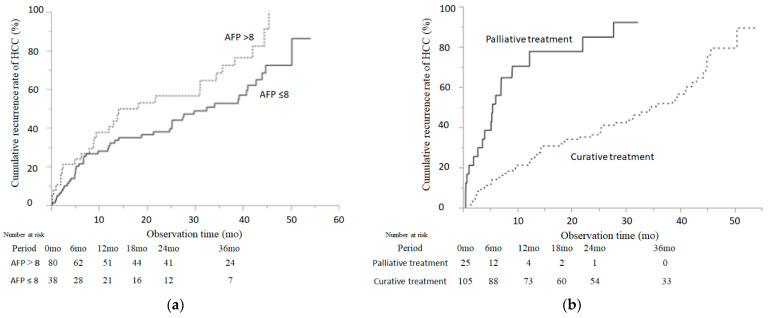
Cumulative incidence of HCC recurrence according to the serum AFP level (AFP) at SVR12 and palliative treatment. Cumulative incidence rates of HCC recurrence are shown by the AFP at SVR12 (**a**) and palliative treatment (**b**). The Kaplan–Meier method and log-rank test were used to assess the cumulative incidence of HCC recurrence. The solid line indicates patients with an AFP at SVR12 ≤ 8 or palliative treatment.

**Figure 3 cancers-13-02257-f003:**
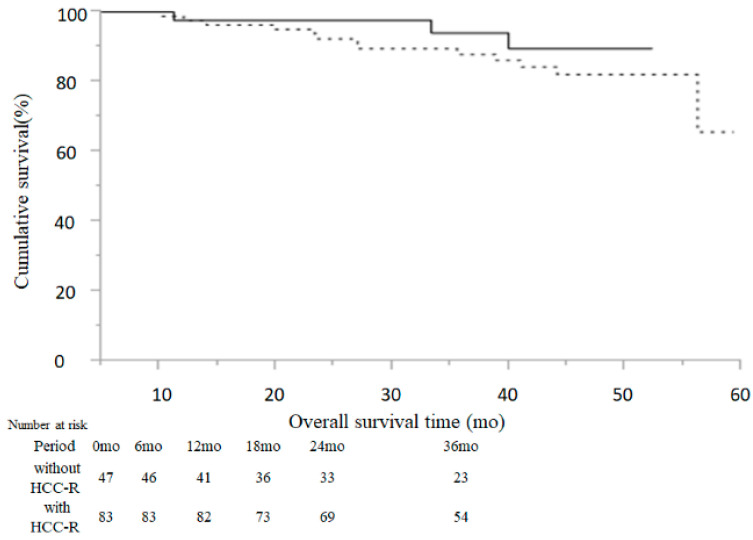
Cumulative rates of overall survival: The Kaplan–Meier method and log-rank test were used to assess the cumulative rates of overall survival. The solid line indicates patients without HCC-R and the dotted line indicates patients with HCC-R.

**Figure 4 cancers-13-02257-f004:**
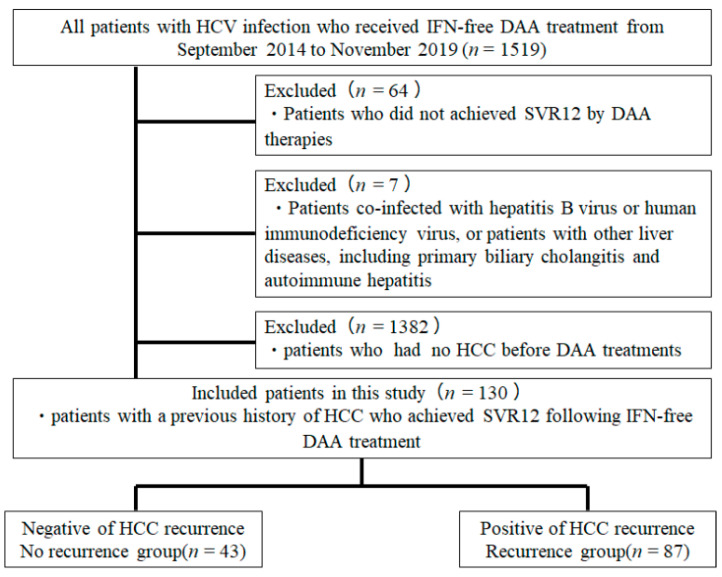
Patient selection criteria.

**Table 1 cancers-13-02257-t001:** Baseline characteristics of the study patients.

Sex (Male/Female)	83 (63.8)/47 (36.2)
Age (years)	75.5 (60–81)
CH/LC	44 (33.8)/86 (66.2)
HCV genotype (1/2 or 3)	98 (75.4)/32 (34.6)
Body mass index (kg/m^2^)	22.9 (20.2–24.9)
Diabetes mellitus (no/yes)	96 (73.8)/34 (26.2)
Treatment history of IFN (no/yes)	97 (74.6)/33 (25.4)
DAA therapy	
ASV +DCV, *n* (%)	22 (16.9)
SOF + LDV, *n* (%)	53 (40.8)
SOF + RBV, *n* (%)	28 (21.5)
ERB + GZR, *n* (%)	8 (6.2)
OBV + PTV + r, *n* (%)	9 (6.9)
GLE + PIB, *n* (%)	10 (7.7)
Number of HCC lesions	1 (1–2)
Size of main tumor lesion (mm)	18 (15–26)
Total number of treatments (times)	1 (1–2)
Final treatment method for HCC before DAA therapy	
Hepatectomy, *n* (%)	23 (17.7)
RFA, *n* (%)	82 (63.1)
TACE, *n* (%)	22 (16.9)
MTA, *n* (%)	3 (1.2)
AST (IU/L)	51(36–65.75)
ALT (IU/L)	39.5 (26–50.75)
Total bilirubin (mg/dL)	0.8 (0.6–1.1)
Albumin (g/dL)	3.7 (3.4–3.9)
Hemoglobin (g/dL)	12.5 (11.5–13.7)
White blood cell count (/μL)	3860 (3100–5010)
Platelet count (×104/μL)	10.0 (7.775–13.7)
AFP (ng/mL)	8.7 (4.2–18)
DCP (mAU/mL)	22 (16.25–33)
Total cholesterol (mg/dL)	149 (131.75–172)
FIB4	5.88 (3.96–8.58)
APRI	1.62 (1.04–2.61)
WFA-M2BP (COI)	4.14 (1.765–6.85)
ALBI Score	−2.48 (−2.73–−2.21)
Child–Pugh score (5/6/7)	89 (68.5)/36 (27.7)/5 (3.8)
HCV-RNA (log copies/mL)	6.0 (5.2–6.4)

Values are median. Values in parentheses are interquartile ranges. Numbers in parentheses are percentage. HCC, hepatocellular carcinoma; CH, chronic hepatitis; LC, liver cirrhosis; HCV, hepatitis C virus; IFN, interferon; DAA, direct-acting antiviral; ASV, asunaprevir; DCV, daclatasvir; LDV, ledipasvir; SOF, sofosbuvir; RBV, ribavirin; GZR, grazoprevir; ERB, elbasvir: OBV, ombitasvir; PTV, paritaprevir; r, ritonavir; GLE, glecaprevir; PIB, pibrentasvir; HCC, hepatocellular carcinoma; RFA, radiofrequency ablation; TACE, transcatheter arterial chemoembolization; MTA, multi-molecular targeted agent; AST, aspartate aminotransferase; ALT, alanine aminotransferase; AFP, α-fetoprotein; DCP, des-γ-carboxy prothrombin; FIB-4, fibrosis-4 index; APRI, aspartate aminotransferase to platelet ratio index; WFA-M2BP, Wisteria floribunda agglutinin-postive Mac-2 binding protein; ALBI, albumin–bilirubin; RNA, ribonucleic acid.

**Table 2 cancers-13-02257-t002:** Baseline characteristics of the study patients between cases with and without HCC-R.

Factor	Cases without HCC-R (*n* = 47)	Cases with HCC-R (*n* = 83)	*p*-Value
Sex (male/female)	26 (55.3)/21 (44.7)	57 (68.7)/26 (31.3)	0.1298
Age (years)	75 (69–82)	76 (67–81)	0.7972
CH/LC	15 (31.9)/32 (68.1)	17 (20.5)/66 (79.5)	0.1460
HCV genotype (1/2 or 3)	32 (68.1)/15 (31.9)	66 (79.5)/17 (20.5)	0.1503
Body mass index (kg/m2)	22.3 (19.9–24.9)	22.8 (20.2–24.9)	0.3085
Diabetes mellitus (no/yes)	37 (78.7)/10 (21.3)	59 (71.1)/24 (28.9)	0.3410
Treatment history of IFN (no/yes)	36 (76.6)/11 (23.4)	22 (26.5)/61 (73.5)	0.6962
DAA regimen (SOF-based DAA/others)			
	25 (53.2)/22 (46.8)	56 (67.5)/27 (32.5)	0.1080
Interval period between the last treatment for HCC and the DAA initiation (months)			
	10.2 (3.7–38.2)	5.8 (2.6–16.3)	0.0288
Number of HCC lesions	1(1–2)	1 (1–2)	0.6757
Size of main tumor lesion (mm)	17 (15–21.5)	18.5 (15–28)	0.0977
Total number of treatments (range)	1 (1–2)	2 (1–2.25)	0.0058
**Final treatment method for HCC before DAA treatment**			
Curative treatment for HCC (i.e., resection + RFA)/palliative treatment (i.e., TACE + MTA)			
	43 (91.5)/4 (8.5)	62 (74.7)/21 (25.3)	0.0208
Hepatectomy, n (%)	11 (23.4)	12 (14.5)	0.3192
RFA, n (%)	32 (68.1)	50 (60.2)	0.5182
TACE, n (%)	3 (6.4)	19 (22.9)	0.0227
MTA, n (%)	1 (2.1)	2 (2.4)	0.9181
**(A) Baseline characteristics**			
AST (IU/L)	51 (34–67)	50 (37–66)	0.2657
ALT (IU/L)	40 (27–57)	39 (26–49)	0.6721
Total bilirubin (mg/dL)	0.8 (0.6–1.0)	0.9 (0.7–1.1)	0.0956
Albumin (g/dL)	3.7 (3.5–3.9)	3.6 (3.3–3.925)	0.8125
Platelet count (×104/μL)	10.9 (7.8–14.6)	9.9 (7.65–12.95)	0.8974
AFP (ng/mL)	8 (3.5–22)	10 (4.775–18)	0.1750
DCP (mAU/mL)	21.34 (17.5–31.5)	23 (16–34.5)	0.2114
Total cholesterol (mg/dL)	149.5 (132–179.25)	148 (128–169)	0.6094
FIB4	5.25 (3.37–7.95)	6.44 (4.19–8.68)	0.0887
APRI	1.62 (0.75–2.49)	1.65 (1.14–2.79)	0.1709
WFA-M2BP (COI)	3.00 (1.72–6.7)	4.27 (1.80–7.51)	0.5039
ALBI score	−2.44 (−2.21–−2.59)	−2.26 (−1.99–−2.65)	0.9183
Child–Pugh score (5/6/7)	31/14/2	58/22/3	0.8976
HCV-RNA (log copies/mL)	6 (5.1−6.35)	6 (5.2−6.5)	0.3110
**(B) Blood test at SVR12**			
AST (IU/L)	25.5 (22–30)	27 (23–36)	0.1629
ALT (IU/L)	16.5 (11–21.75)	19 (13–23)	0.6251
Total bilirubin (mg/dL)	0.65 (0.5–1.0)	0.9 (0.6–1.225)	0.0157
Albumin (g/dL)	4.0 (3.725–4.3)	3.9 (3.675–4.2)	0.1312
Platelet count (×104/μL)	11.85 (9.7–14.8)	10.55 (7.975–14.7)	0.7872
AFP (ng/mL)	5 (3–7)	6.2 (3.925–9.625)	0.0199
DCP (mAU/mL)	20 (17–29)	25 (16–45)	0.2982
Total cholesterol (mg/dL)	183 (156.5–209.5)	169.5 (133–193.25)	0.0151
FIB4	4.14 (3.21–5.42)	4.73 (3.25–6.35)	0.0970
APRI	0.73 (0.53–0.96)	0.90 (0.54–1.37)	0.0724
WFA-M2BP (COI)	2.63 (1.56–5.63)	3.27 (1.58–6.12)	0.5169
ALBI score	−2.71 (−2.50–−2.98)	−2.61 (−2.30–−2.81)	0.0322
Child–Pugh score (5/6/7)	42/5/1	72/10/1	0.8930

Values are medians. Values in parentheses are interquartile ranges. Numbers in parentheses are percentages. HCC, hepatocellular carcinoma; R, recurrence; CH, chronic hepatitis; LC, liver cirrhosis; HCV, hepatitis C virus; IFN, interferon; DAA, direct-acting antiviral; ASV, asunaprevir; DCV, daclatasvir; LDV, ledipasvir; SOF, sofosbuvir; RBV, ribavirin; GZR, grazoprevir; ERB, elbasvir: OBV, ombitasvir; PTV, paritaprevir; r, ritonavir; GLE, glecaprevir; PIB, pibrentasvir; RFA, radiofrequency ablation; TACE, transcatheter arterial chemoembolization; MTA, multi-molecular targeted agent; AST, aspartate aminotransferase; ALT, alanine aminotransferase; AFP, α-fetoprotein; DCP, des-γ-carboxy prothrombin; FIB-4, fibrosis-4 index; APRI, aspartate aminotransferase-to-platelet ratio index; WFA-M2BP, Wisteria floribunda agglutinin-positive Mac-2 binding protein; ALBI, albumin–bilirubin; RNA, ribonucleic acid.

**Table 3 cancers-13-02257-t003:** Independent predictors for HCC-R.

Factor	Category	Multivariate Analysis		
		Hazzard Ratio	95%CI	*p*-Value
Interval period between the last treatment for HCC and DAA initiation				
	For every 1 month	0.999	0.992–1.001	0.5183
Total number of treatments	For each time	1.028	0.834–1.229	0.7824
Final treatment method for HCC before DAA treatment				
Curative treatment (i.e., resection + RFA)/ palliative treatment (i.e., TACE + MTA)	Curative versus palliative	3.974	1.924–8.207	0.0006
**Blood test at SVR12**				
Total bilirubin	For every 1 mg/dL	1.051	0.550–1.914	0.8762
AFP	For every 1 ng/mL	1.048	1.016–1.077	0.0047
Total cholesterol	For every 1 mg/dL	0.996	0.987–1.004	0.3147
ALBI score	For every 1	1.601	0.602–4.024	0.8930

Values are medians. Values in parentheses are interquartile ranges. HCV, hepatitis C virus; IFN, interferon; DAA, direct-acting antiviral; TACE, transcatheter arterial chemoembolization; MTA, multi-molecular targeted agent; AST, aspartate aminotransferase; AFP, α-fetoprotein; ALBI, albumin–bilirubin.

**Table 4 cancers-13-02257-t004:** Relative changes in HCC status and serological data at baseline and at HCC-R after DAA treatment.

Factor	At Baseline	At HCC-R after DAA Treatment	p-Value
Number of HCC lesions	1 (1–2)	1 (1–2)	0.2378
Size of main tumor lesion (mm)	18.5 (15–28)	15 (12–20)	0.0138
Barcelona-Clinic Liver Cancer staging classification(0/A/B/C)	35/22/22/3	33/28/18/3	0.7581
AST (IU/L)	50 (37–66)	26 (23–36)	<0.001
ALT (IU/L)	39 (26–49)	16 (14–24)	<0.001
Total bilirubin (mg/dL)	0.9 (0.7–1.1)	0.9 (0.6–1.0)	0.3895
Albumin (g/dL)	3.6 (3.3–3.925)	4.1 (3.6–4.3)	<0.001
Platelet count (×104/μL)	9.9 (7.65–12.95)	11.4 (9.5–14.7)	0.0032
AFP (ng/mL)	10 (4.775–18)	9 (5–33.25)	0.0459
DCP (mAU/mL)	23 (16–34.5)	36.5 (19.75–139.25)	0.1709
ALBI score	−2.26 (−2.81–−1.96)	−2.70 (−2.94–−2.28)	<0.001
Child–Pugh score (5/6/7)	58/22/3	72/10/1	0.0301

Values are medians. Values in parentheses are interquartile ranges. HCC, hepatocellular carcinoma; AST, aspartate aminotransferase; ALT, alanine aminotransferase; AFP, α-fetoprotein; DCP, des-γ-carboxy prothrombin; ALBI, albumin–bilirubin.

## Data Availability

The data used in the present study are available from the corresponding author upon reasonable request.

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
