# Peer review of "Long-Term Outcomes and Evaluation of Hepatocellular Carcinoma Recurrence after Hepatitis C Virus Eradication by Direct-Acting Antiviral Treatment: All Kagawa Liver Disease Group (AKLDG) Study"

_cancers, 2021, doi:10.3390/cancers13092257_

Round 1
Reviewer 1 Report
- Multivariate analysis showed that transcatheter arterial chemoembolization TACE
before DAAs were associated with independent factors for HCC recurrence.
It is interesting date.
Author should discuss about the reason in discussion.
- In Table 3,is it correct (Odds ratio)→(Hazzard ratio)?
Reviewer 2 Report
There are conflicting reports about the long-term outcomes of patients with hepatocellular carcinoma (HCC) recurrence after direct-acting antiviral (DAA) treatment. In this retrospective study the Authors aimed to investigate recurrence rates, recurrence factors, and prognosis of 130 patients treated with DAAs after HCC treatment. The median observation time was 41±13.9 months after DAA treatment. The recurrence rates of HCC were 23.2%, 32.5%, 46.3%, and 59.4% at 6, 12, 24, and 36 months, 45 respectively. Multivariate analysis showed that transcatheter arterial chemoembolization (TACE) before DAAs (HR=3.523; 95% CI 1.611–7.255; P=0.0023) and Alpha-fetoprotein at sustained virological response (HR=1.045; 95% CI 1.011–1.076; P=0.0133) were associated with independent factors for HCC recurrence. The 12-, 24-, and 36-month overall survival rates were 97.6%, 94.0%, and 89.8%, 49 respectively. The 12-, 24-, and 36-month survival rates of the no-recurrence and recurrence groups were 97.7%, 97.7%, and 94.1%, and 97.6%, 94.0%, and 89.8%, respectively (P=0.3404).
Interestingly, both size of the main tumor lesion and the serological data were significantly improved at the time of HCC recurrence after DAA treatment.
They concluded that their results show an improved prognosis regardless of recurrence rate, which suggests that DAA treatment in HCV patients should be considered.
The study is of interest and the topic of clinical relevance. However, some points deserve further details.
-Patients’ characteristics: please describe the study design regarding the follow-up of patients following antiviral therapy. Underwent they surveillance (ultrasound or other imaging)?
-Table 3: please add also the Child-Pugh score as this is the most used tool to assess liver function. Moreover, it would be relevant to add the Child-Pugh score of patients at the time of HCC recurrence since it affects treatment allocation and overall survival, as reported in a comprehensive recent review (Non-transplant therapies for patients with hepatocellular carcinoma and Child-Pugh-Turcotte class B cirrhosis. The Lancet Oncology 2017 (18); 2: e101-e112) and antiviral therapy can avoid liver function deterioration which in turns limit HCC treatments.
-To further improve the Discussion I would suggest discussing a pivotal study where a meta-analysis showed that recurrence risk and survival are extremely variable in patients with successfully treated HCV-related HCC, providing a useful benchmark for indirect comparisons of the benefits of DAAs and for a correct design of randomized controlled trials in the adjuvant setting (A meta-analysis of single HCV-untreated arm of studies evaluating outcomes after curative treatments of HCV-related hepatocellular carcinoma. Liver International 2017;37 (8):1157-1166.
- Due to the clinically relevant results of the study I would suggest emphasizing in the Discussion the potential role of DAA therapy discussing literature data demonstrating how the improved viral liver disease treatment impact survival (The evolutionary scenario of hepatocellular carcinoma in Italy: an update. Liver International 2017; 37(2):259-270.
Reviewer 3 Report
To Authors
The data reported in the paper of Tani et al. are not original, but they add relevance to the already known concept that liver cancer has a well-established clinical history, where the tumour presentation and treatment characteristics determine the disease prognosis, but the novel aspect of this paper is that DAA therapy seems to ameliorate prognosis.
Major suggestions:
Table 2: must be modified, please use “Cases without HCC-R” and “Cases with HCC-R” and report number (and percentage), also for all factors described in the table. Please, consider to subgroup and compare between 2 groups: cases with HCV-1 versus cases with HCV non-1 and the same between cases treated with Sofosbuvir-based DAA therapy versus other schedules used. This representation could better clarify the power of the p-value shown in some cases treated with Sofosbuvir. For the same reason, I would like to propose the comparison also between cases that received a curative therapy for HCC (i.e., resection+RFA) and cases that received a palliative therapy (i.e., TACE+MTA). Probably this manner of grouping treatment strategies will emphasize results.
Paragraph 2.4, please consider that here univariate analysis is repetitive of the data obtained in the comparison shown in table 2 by p-values, and the multivariate analysis used must be better specified and must be reconsidered with the subdivision proposed in table 2. Thus I proposed to show only the data obtained by multivariate analysis in table 3.
Paragraph 2.5, please explain more accurately this paragraph, it is not clear, particularly on line 169 … at SVR12 before DAA treatment ?? what does it mean ?? Paragraph 2.6, probably here Figure 2 side b) must be modified according to numbers of cases that received palliative therapy (TACE+MTA), as suggested in table 2.
Paragraph 2.7, I realized, that the deceased cases have all been in the group of cases with HCC-R. Is that true? Or perhaps is there an uncorrected figures description between OS and cases with HCC-R survival rates, as these appeared equal at 12, 24 and 36 mos, and both reported as 97.6%, 94% and 89.8% ??? Please here also specify who many cases died in the groups of cases without and with HCC-R.
Then in figure 3, only figure side b) can be kept and side a) deleted. Please also report below the figure the cases at risk among those with and without HCC-R at each time-point (i.e., at 10, 20, 30 mos. and so on), as you did in figure 2.
Paragraph 2.8, please specify that this subanalysis regards the 87 cases with HCC-R compared by factors at 2 time-points that were defined “At baseline” and “At HCC-R after DAA” (see also in minor revisions: Line 194 and Table 4).
Discussion must be extensively revised !! Particularly, the Authors should discuss PROs and CONTs by data showed, that lead to the conclusions on the ameliorate prognosis in the cohort of patients with a previous history of HCC after DAA therapy, as 83 cases had HCC-R during FU after DAA and 16 cases dead, especially among HCC-R group. Please make the discussion more fluent in the English language.
Conclusions must be ameliorated !!
Minor suggestions:
Please, change all over through the manuscript “HCV-serotype” with “HCV-genotype”
Table 1: please add to numbers (percentages) for every line in parameters considered, specify numbers (percentages) of genotype HCV-1, HCV-2 and HCV-3. Here is possible to know the stage (number and percentages) of liver fibrosis of cases with CH (i.e., F2, F3) and Child score for cases with LC (staged F4)
Line 194, after…data before DAA…, please specify adding “at baseline”
Line 200, change … with before DAA treatment … “with the same figures at baseline”
Table 4: Factors: “At baseline” versus “At HCC-R after DAA”
Figure 4: please must be redone and magnificate, because it is not legible
Paragraph 4.1, Line 334, exclusion criteria were: patients coinfected (HBV or/and HIV) or with history of other liver diseases (PBC, PSC, others)……, while inclusion criteria were: patients with SVR12 and with history of previous HCC treated and cured before DAA initiation. Please specify the time (mos+SD) from the last curative HCC treatment and the DAA initiation.
Paragraph 4.3, please describe better the type of statistic employed for categorical and continuous variables analysis and specify…. what type of multivariate analysis was performed and if it was applied only for variables identified by univariate analysis with a p-value <0.1 or p<0.05 ??
Round 2
Reviewer 2 Report
The Authors satisfactorily addressed the raised points and now the manuscript can be accepted.
Reviewer 3 Report
To Authors
The paper of Tani J. et al., titled “Long-term outcomes and evaluation of HCC recurrence after hepatitis C virus eradication by DAA: the AKLDG study” concerns the evaluation of incidence, characteristics and predictors of HCC recurrence (HCC-R) in a large cohort of cases treated with DAA for chronic HCV infection. One-hundred and 30 cases were analysed, of which 83 (64%) had HCC recurrence during a FU of 41+14 mos. Authors demonstrated that monitoring AFP levels can be useful to early identify patients with high risk of HCC-R, particularly those that received TACE and were treated with DAA too close from the last HCC cure.
The paper in the present form should again be revised before being accepted for publication in Cancers. Unfortunatly the Authors have not well arranged the paper, in particular the discussion, that must still be revised to be shorten and make more fluid and conclusions that will be to extensively revised. Furthermore, I propose the following changes:
Simply summary: from line 36 to 40The recurrence rate of HCC was relatively high and corresponding to 63.8% (83/130) of cases. These patients electively received a palliative treatment before the DAA therapy and showed a shorter interval period between the last treatment for HCC and the DAA initiation, also displaying a significant increase of alpha-fetoprotein during follow-up after therapy, respect to cases without HCC-R. Overall survival was comparable in both groups, because of the improvement in liver function tests after the DAA therapy, thus no differences in survival rates were observed between patients with or without HCC-R. The results of this study indicate that IFN-free DAA treatment after HCC could be recommended to improve prognosis in this subset of cases. However, it remains imperative to observe the timing, at least 9-12 months after the last treatment for HCC, before proposing therapy with DAA.
Table 1, line 100, please correct “HCV genotype (1/2 or 3)”
Line 191-200, it is unclear!? but I suppose that this description regards the old table 3 where univariate analysis appared together with multivariate, thus the p values are not the same reported in the actual table 2, where there is the comparison between cases without and with HCC-R. Please provide the specific changes in the paragraph. For example ALBI score does not appear significant ??! Please be careful that even in the discussion (line 288-293) there are incorrect data, still set on the old table 3.
Table 3, please differentiate “Factors” as. A) Baseline characteristics (in bold) and B) Blood test at SVR12 (in bold). Delete “Blood test at DAA administration”.Line 294-299, …AFP level is not a “risk factor” but a diagnostic marker for HCC
Please reconstruct the paragraph from line 294 to 299 “Elevated AFP levels are recognized as one of the common diagnostic marker for HCC, particularly when >200 ng/mL [30,31]. It is known that elimination of the virus by IFN significantly decreases AFP levels [31,32]. Thus, in our study AFP at pre-treatment did not appeared significantly different between cases without and with HCC-R, but at post-treatment evaluation this marker became an independent predictor of carcinogenesis, as also described in Literature [33], being at cut-off values >8.0 ng/mL statistically associated to HCC-R”
Line 317-321, please simplify as: “Therefore, 83 cases had HCC-R after the DAA treatment, but these cases did not show the worsening of tumor-specific factors (such as number, size, and BCLC staging classification), of tumor markers (such as AFP and DCP) or of liver functional indexes (such as AST, ALT, bilirubin), and of CPT score, on the contrary, they appeared to have improved from baseline”.
Line 323, … “before to after DAA treatment”.
Line 325, delete the sentence: “Thus, DAA treatment may be associated with the improvement or preservation of liver function”.
Line 357, … is significant by examination of a large number of patients treated and by a real clinical setting.
Finally, I ask the Authors if they are interested in publishing their manuscript in the special issue on HCV&HCC, at the following site: Cancers | Special Issue : Hepatitis C Virus and Hepatocellular Carcinoma (HCV & HCC) (mdpi.com), since the topic is precisely focused on the their study and the dead-line has been postponed to 30 of september. If you are interested, on this, please make this writte on the cover letter with the revised paper at the act of the last submission.
